# Isolation, Identification and Pharmacological Effects of *Mandragora autumnalis* Fruit Flavonoids Fraction

**DOI:** 10.3390/molecules27031046

**Published:** 2022-02-03

**Authors:** Nawaf Al-Maharik, Nidal Jaradat, Najlaa Bassalat, Mohammed Hawash, Hilal Zaid

**Affiliations:** 1Division of Chemistry, Faculty of Science, An-Najah National University, Nablus 00970, Palestine; 2Department of Pharmacy, Faculty of Medicine and Health Sciences, An-Najah National University, Nablus 00970, Palestine; mohawash@najah.edu; 3Department of Biology and Biotechnology, Faculty of Sciences, Arab American University, Jenin P.O. Box 240, Palestine; nbassalat@gmail.com; 4Qasemi Research Center, Faculty of Medicine, Al-Qasemi Academic College, Arab American University, Jenin P.O. Box 240, Palestine; hilal.zaid@gmail.com

**Keywords:** *Mandragora autumnalis*, flavonoids, cytotoxicity, antidiabetic, anti-obesity, DPPH scavenging, antimicrobial

## Abstract

Since ancient times, *Mandragora autumnalis* has been used as a traditional medicinal plant for the treatment of numerous ailments. In light of this, the current study was designed to isolate and identify the chemical constituents of the flavonoids fraction from *M. autumnalis* ripe fruit (FFM), and evaluate its DPPH scavenging, anti-lipase, cytotoxicity, antimicrobial and antidiabetic effects. An ethyl acetate extract of *M. autumnalis* was subjected to a sequence of silica gel column chromatography using different eluents with various polarities. The chemical structures of the isolated compounds were identified using different spectral techniques, including ^1^H NMR and ^13^C NMR. FFM’s anti-diabetic activity was assessed using a glucose transporter-4 (GLUT4) translocation assay, as well as an inhibition against α-amylase and α-glucosidase using standard biochemical assays. The FFM anti-lipase effect against porcine pancreatic lipase was also evaluated. Moreover, FFM free radical scavenging activity using the DPPH test and antimicrobial properties against eight microbial strains using the micro-dilution method were also assessed. Four flavonoid aglycones were separated from FFM and their chemical structures were identified. The structures of the isolated compounds were established as kaempferol **1**, luteolin **2**, myricetin **3** and (+)-taxifolin **4**, based on NMR spectroscopic analyses. The cytotoxicity test results showed high cell viability (at least 90%) for up to 1 mg/mL concentration of FFM, which is considered to be safe. A dose-dependent increase in GLUT4 translocation was significantly shown (*p* < 0.05) when the muscle cells were treated with FFM up to 0.5 mg/mL. Moreover, FFM revealed potent *α*-amylase, *α*-glucosidase, DPPH scavenging and porcine pancreatic lipase inhibitory activities compared with the positive controls, with IC_50_ values of 72.44 ± 0.89, 39.81 ± 0.74, 5.37 ± 0.41 and 39.81 ± 1.23 µg/mL, respectively. In addition, FFM inhibited the growth of all of the tested bacterial and fungal strains and showed the greatest antibacterial activity against the *K. pneumoniae* strain with a MIC value of 0.135 µg/mL. The four flavonoid molecules that constitute the FFM have been shown to have medicinal promise. Further in vivo testing and formulation design are needed to corroborate these findings, which are integral to the pharmaceutical and food supplement industries.

## 1. Introduction

Humankind has been fascinated with botanicals since antiquity because of their exceptional nutritional, cosmetic and medicinal characteristics [1]. A growing body of evidence shows that herbal medicines have fewer side effects than most synthetic treatments, making them an important source for the development of new pharmaceuticals, cosmetics and nutritional supplements [2]. It is a controversial subject in the medical and pharmaceutical sectors to study natural products for their pharmacological properties [3]. Consequently, biological and chemical screenings of wild herbs are essential in the creation of new medicinal and functional food products.

*Mandragora autumnalis* Bertol. (also known as *Mandragora officinarum* L.) is a highly valued wild medicinal herb. It is a perennial herbaceous plant in the Solanaceae family with purple or violet blooms and orange or yellow mature fruits (berries). It is widely distributed throughout the Mediterranean Basin, including Palestine. It is still used in traditional Arabian medicine, especially Palestinian herbal medicine, to treat pains, insomnia, cough, throat pain, bronchitis and various genital organs diseases [4,5]. *M. autumnalis* has antitumor [6], antioxidant and antimicrobial activities [7], as well as having free radicals, cholinesterase, tyrosinase, α-amylase and α-glucosidase inhibitory activities [8].

Due to the tendency of unripe fruits, leaves and roots to cause mental disorder and arouse the senses, it is known locally as Tufah Almajan (Satan’s Apple), love apples and Beed Aljin (Eggs of the Jinn) [9]. Historically, inhabitants of the Islamic Empire used *M. autumnalis* dried ripe berries to treat metabolic risk factors such as diabetes and obesity, while its roots and leaves were used to treat a variety of illnesses, such as skin ulcers, infected wounds, pimples, warts, mouth inflammation, eye infections, pain, vomiting, male infertility and insomnia [10,11,12]. Previous phytochemical studies of *M. autumnalis* resulted in the isolation of 11 alkaloids, including calystegine A3 **5**, scopine **6** and a few coumarins such as scopoletin (4-methylesculetin) **7** as well as the steroid sitosterol **8** in various parts of *M. autumnalis* roots (Figure 1), fruits and leaves [11,13]. Furthermore, several withanolides (polyoxygenated steroidal lactones/lactols) such as mandragorolide A **9** were isolated from the *M. autumnalis* whole plant methanolic extract [14]. Flavonoids have never been extracted from any part of this plant, according to a review of the literature.

Noncommunicable diseases, including obesity, diabetes, oxidative stress and cancer, are considered among the most virulent diseases of the 21st century. The combined burden of these diseases is rising fastest in communities where they impose large, unavoidable costs in human, social and economic terms [15].

The development of drug-resistant pathogens has occurred rapidly, while the emergence of multi-drug resistant strains has increased exponentially in recent years [16].

Despite a variety of measures taken in recent decades to address this problem, worldwide antimicrobial resistance trends show no indications of abating. One of the primary causes of the rise of antimicrobial resistance is the overuse and misuse of antimicrobial agents in both the healthcare and agricultural sectors. Microbial mutation and spontaneous evolution, as well as horizontal gene transfer that passes resistant genes on, all play important roles in the rise of antimicrobial resistance [17].

Following up on a previously conducted study that involved the identification of phytochemical composition and the assessment of biological activities of various wild-growing herbs in Palestine, we have considered investigating the ethyl acetate fraction of the methanol extract of *M. autumnalis* ripe berries, identifying the chemical composition of this fraction and assessing its anti-obesity, DPPH scavenging, cytotoxic, antimicrobial and antidiabetic effect by measuring its action on the glucose transporter-4 (GLUT4) translocase, α-amylase and α-glucosidase properties.

## 2. Results

### 2.1. Chemistry

#### Identification of the Isolated Compounds

Four flavonoids (Figure 2) were isolated from the ethyl acetate (EtOAc) extract of the ripe fruit of *M. autumnalis*, and their structures were identified as kaempferol **1**, luteolin **2**, myricetin **3** and (+)-taxifolin (**4**) by comparing their ^1^H- and ^13^C-NMR spectral analysis with that reported in the literature (Table 1, Appendix A).

### 2.2. Biological Evaluations

#### 2.2.1. Cytotoxicity

Following MTT assays, the toxicity of FFM was assessed in vitro in the L6-GLUT4 myc cell. Concentrations of FFM that preserved at least 90% cell viability were deemed to be safe. FFM was proven to be safe at concentrations of up to 1 mg/mL. The cells’ viability was the lowest at 1 mg/mL and reached 93 ± 2%. As a result, the effectiveness trials were carried out at lower dosages of up to 0.5 mg/mL.

#### 2.2.2. GLUT4 Translocation to the Plasma Membrane

Increasing GLUT4 translocation to the plasma membrane (PM) in muscle is an important action of insulin for maintaining glucose homeostasis. As expected, insulin enhanced GLUT4 translocation to the surface of myoblasts and thus increased glucose uptake. To examine the effect of FFM on GLUT4 translocation to the plasma membrane in the absence and presence of insulin, L6-GLUT4myc, the cells were exposed to increased concentrations of FFM for 20 h. The effect of FFM was compared with that of the control and insulin (100 nM)-treated cells. Insulin-stimulated GLUT4 translocation to the PM by 1.66 ± 0.25 folds compared with vehicle untreated cells. A dose-dependent increase in GLUT4 translocation was significantly noticed (*p* < 0.05) when the muscle cells were treated with FFM in the absence and the presence of insulin, except at 0.125 mg/mL. The FFM at concentrations of 0.125, 0.25 and 0.5 µg/mL significantly enhanced GLUT4 translocation (*p* < 0.05) at no insulin treatment by 1.33 ± 0.13, 1.33 ± 0.22 and 1.56 ± 0.21 µg/mL, respectively. The FFM also stimulated GLUT4 translocation to the PM in the presence of insulin significantly at 0.25 and 0.5 mg/mL by 2.35 ± 0.43 and 2.35 ± 0.19, respectively (Figure 3).

#### 2.2.3. α-Amylase Inhibitory Potential

The results revealed that FFM displayed *α*-amylase inhibitory activity in a dose-dependent manner compared with the commercial antidiabetic therapeutic agent acarbose, as indicated in Figure 4. However, the IC_50_ of FFM was 72.44 ± 0.89 µg/mL, while the IC_50_ of the positive control acarbose was 28.18 ± 1.22 µg/mL.

#### 2.2.4. α-Glucosidase Inhibitory Activity

The α-glucosidase inhibitory activity of FFM was assessed and compared with that of the strong α-glucosidase inhibitory agent (acarbose). In Figure 5, the α-glucosidase inhibitory activity of FFM and acarbose was shown, and the IC_50_ of FFM against this enzyme was 39.81 ± 0.74 µg/mL, which is considered very close to the IC_50_ of the positive control acarbose, which was 37.15 ± 0.33 µg/mL.

#### 2.2.5. DPPH Scavenging Activity

The ability of FFM to scavenge free oxygen radicals was assessed using the DPPH method, while trolox (vitamin E analog) was used as a positive control. The DPPH scavenging activity of FFM and trolox is shown in Figure 6, with the IC_50_ of FFM being 5.37 ± 0.41 µg/mL, which is considered close to that of trolox (IC_50_ = 2.23 ± 1.23 µg/mL).

#### 2.2.6. Antilipase Activity

The hydrolysis of *p*-nitrophenyl butyrate to *p*-nitrophenol was used to measure the influence of FFM on the porcine pancreatic lipase enzyme. The assay detected the inhibitory activity of FFM compared with orlistat, a strong lipase inhibitory agent. The results of the lipase enzyme inhibitory activity are shown in Figure 7, and the IC_50_ of FFM against this enzyme was 39.81 ± 1.23 µg/mL, while the IC_50_ of the positive control orlistat was 12.56 ± 0.35 µg/mL.

#### 2.2.7. Antimicrobial Capacity

The microdilution assay was used to evaluate the antimicrobial activity of the FFM against MRSA, *S. aureus*, *E. faecium*, *S. sonnie*, *P. aeruginosa*, *E. coli*, *E. floccosum* and *C. albicans*. The results revealed that FFM inhibited the growth of all of the tested Gram-positive and Gram-negative microbial strains, as indicated in Table 2.

## 3. Discussion

Plant extracts have been utilized in medicine, food preservation, medicines and cosmetics for hundreds of years. Since the time of Hippocrates (460–377 BC), they have been used to treat a wide range of human and animal ailments, including infectious, systemic and inflammatory ones [18]. The primary focus of many pharmaceutical experts is the discovery and screening of possible anti-diabetic, anti-obesity, antioxidant and antibacterial medications derived from natural ingredients. Flavonoids are a broad class of polyphenols that are produced by plants and are considered secondary metabolic products. They are chemically categorized into flavonols, flavones, flavanones, chalcones, isoflavones, aurones, anthocyanidins, flavan-3,4-diols and flavan-3-ols subgroups, each with its own set of physicochemical qualities, biological features and structural characteristics [19].

### 3.1. Chemical Characterization

Flavonoids are the primary active components of several therapeutic plants and foods, exerting a wide variety of pharmacological effects [20]. *M. autumnalis* has a long history of medicinal usage in the Arab world. Twenty-two alkaloids have previously been extracted and identified from the roots, leaves and flowers of *M. autumnalis*, while 55 chemicals representing the volatile components of *M. autumnalis* have been identified using capillary GC-MS [21]. Additionally, five coumarins have been isolated and identified from *M. autumnalis* roots, leaves and unripe fruits in Jordan Valley [11]. To the best of the authors’ knowledge, no previous study has been conducted on the phytochemical components of *M. autumnalis* ripe fruits. Thus, the purpose of this work was to separate polyphenols, particularly flavonoids, from mature fruits of *M. autumnalis* methanolic extract. Four flavonoids (one flavones **2**, two flavanols **1** and **3** and one flavanonol **4**) were isolated utilizing a sequential flash column chromatography and preparative TLC.

Flavonoid **1** was obtained as a yellow powder with the molecular formula of C_15_H_10_O_6_ as shown from ESI-HRMS [M + H]^+^ at *m*/*z* 287.0556. The ^13^C-NMR spectrum of **1** displayed resonance for all of the 15 carbons present in the molecule as indicated in the ^13^C-NMR (DEPT) (Appendix A). The ^1^H- and ^13^C-NMR spectrum (Table 1, Appendix A) of **1** were similar to those reported in the literature for kaempferol [22,23]. Compound **2** was obtained as a pale yellow powder with a molecular peak at *m*/*z* = 287.0447 [M + H]^+^, suggesting that its molecular formula was C_15_H_10_O_6_. The ^1^H- and ^13^C-NMR spectra validated the skeleton as the luteolin **2** framework (Appendix A) due to the presence of three ABX-type aromatic protons at 7.43 (1 H, dd, *J* = 8.2, 2.2 Hz, H-6’), 7.40 (1 H, d, *J* = 2.2 Hz, H-2’) and 6.89 (1 H, d, *J* = 8.2 Hz, H5’), as well as the presence of four OH peaks. The HSQC and HMBC experiments supported the structure of **2** [20,22]. Thus, the structure of **2** was identified as 5,7,3′,4′-tetrahydroxyflavone (luteolin). Compound **3** was also obtained as a yellow amorphous powder. Its HR-ESI-MS spectra revealed a molecular peak at *m*/*z* = 319.0476 [M + H]^+^, suggesting that it had the chemical formula C_15_H_10_O_8_. The ^1^H-NMR spectra revealed that **3** was a flavonol with six OH peaks and four aromatic proton peaks (Appendix A). When ^1^H- and ^13^C-NMR (Table 1, Appendix A) data were compared with published data [22,24], the chemical structure of **3** was discovered to be a myricetin skeleton.

Compound **4** was isolated as a pale yellow solid and its chemical formula was established as C_15_H_12_O_7_ based on ESI-HRMS *m*/*z* 305.0531 [M + H]^+^ and NMR spectral data. Compound **4** had 15 carbon atoms, including 8 quaternary carbons and 7 methine carbons, according to the DEPT and HMQC spectra (Appendix A). The presence of flavanonol framework was evident from the ^1^H NMR spectrum (Table 1, Appendix A), which displayed the presence of one-proton doublet of a doublet at 4.51 (*J* = 11.2, 5.9 Hz) and a doublet at 4.99 (*J* = 11.2 Hz) corresponding to H-3 and H-2, respectively. The carbon resonances at 72.0, 83.5 and 198.3, designated C-3, C-2 and C-4 of the flavanonol C-ring, respectively, confirmed this identification. H-2 and H-3 were trans, as shown by their coupling constants of 11.2 Hz. The ^1^H-NMR spectrum (Appendix A) revealed the existence of aromatic signals at 5.89 and 5.94, corresponding to H-6 and H-8, respectively, and a multiplet in the area 6.74, corresponding to H-5’,6’. ^1^H- and ^13^C-NMR spectra, further revealed the presence of five hydroxyl groups, of which the 3-OH hydroxyl resonated at *δ*_H_ 5.74. In the HMQC spectrum (Appendix A), the proton signal at 4.51 (H-3) showed one bond correlation with the carbon signal at 72.0. The proton signal at *δ*_H_ 4.99 (H-2) had long-range correlations with carbon signals at *δ*_C_ 128.5 and 72.0, showing carbon signals at *δ*_C_ 128.5 and 72.0 to be due to C-1′ and C-3, respectively (Appendix A). All of these data matched those reported in the literature, confirming the identity of compound **4** as taxifolin [25].

### 3.2. Antidiabetic Effects

Diabetes mellitus is a metabolic disorder characterized by the failure of insulin secretion by pancreatic *β*-cells, or poor responses of cells to insulin. Diabetes could lead to long-term complications such as microangiopathy, neuropathy and retinopathy. Acarbose, a synthetic antidiabetic used to suppress the activity of *α*-amylase and *α*-glucosidase in the small intestine, is linked with a number of adverse effects, including intestinal pneumatosis, abdominal distention, gas and diarrhea [26]. Additionally, some synthetic antidiabetic medications have severe adverse, and in some cases fatal, side effects. As a result, scientists have focused their research during the last two decades on natural products such as flavonoids as potential substitutes for acarbose and other antidiabetic medications.

#### 3.2.1. FFM Effects on GLUT4 Translocation

Normal glucose metabolism, crucial in maintaining homeostasis, is regulated by two primary hormones, namely insulin and glucagon. Following hydrolysis of starch in the upper gastrointestinal tract, the formed monosaccharides are absorbed through glucose transporters (GLUT) into the bloodstream. Fourteen GLUT proteins, including GLUT 1-12, myoinositol transporter (HMIT) and GLUT 14, are expressed in humans [27]. Both glucose and insulin are potent enhancers of hippocampal memory processes and it has been suggested that insulin’s cognitive effects may occur via upregulation in GLUT4-mediated glucose uptake [28]. Indeed, GLUT4 is responsible for the majority of the circulating blood glucose clearance and uptake into skeletal muscle, liver and fat cells. GLUT4 is insulin-sensitive and upon insulin stimulation, GLUT4 is translocated to the PM from intracellular vesicular [28]. However, this process is diminished in insulin resistance and diabetes type 2.

The effect of FFM on GLUT4 translocation to the PM in the L6 skeletal muscle cell line was examined in this work. The results indicate that FFM dramatically increased the amount of GLUT4 in the PM in both the presence and the absence of insulin. These findings suggest that FFM may include anti-diabetic chemicals. Flavonoids have been shown to have anti-diabetic action via distinct mechanisms. Certain flavonoids have been shown to enhance glucose uptake by hepatocytes and myocytes [29]. The stimulatory effect of the four flavonoids mixture on GLUT4 translocation reported here is attributed to kaempferol and myricetin. Both have been shown to improve glucose elimination in diabetic rats fed a high-fat diet [30,31].

Most notably, kaempferol reduced GLUT4 and AMPK expression in muscle and adipose tissue in obese mice while reversing the high-fat diet. In obese diabetic mice, kaempferol therapy improved hyperglycemia and glucose tolerance [29]. These findings, together with our own, imply that kaempferol’s anti-diabetic action is mediated in part by increased GLUT4 translocation and activation.

#### 3.2.2. Carbohydrates Metabolic Enzymes

The findings of this study demonstrated that FFM has potent *α*-amylase and *α*-glucosidase activity. The FFM displayed potent *α*-amylase inhibitory action with an IC_50_ value of 72.44 ± 0.89 µg/mL, lower than the commercial anti-diabetic medication acarbose (IC_50_ = 28.18 ± 1.22 µg/mL). These findings are consistent with those published earlier by Proenca et al., who used a microanalysis screening method to examine the capacity of 40 structurally similar flavonoids to inhibit *α*-amylase activity [32]. It was reported that kaempferol and luteolin and myricetin inhibited *α*-amylase with an IC_50_ value of 118 ± 7, 78 ± 3 and 107 ± 6 µM. Moreover, this plant fraction displayed powerful *α*-glucosidase inhibitory activity with an IC_50_ value of 39.81 ± 0.74 µg/mL, which is comparable to that of acarbose (IC_50_ = 37.15 ± 0.33 µg/mL).

These findings add to prior research indicating that natural polyphenols, particularly flavonoids, can inhibit the activity of carbohydrate-hydrolyzing enzymes such as *α*-amylase and *α*-glucosidase [33]. A review article concerning pharmacokinetic effects and the structure-activity relationship (SAR) between flavonoids and *α*-amylase and *α*-glucosidase inhibitory activities highlighted that various flavonoids displayed higher *α*-glucosidase inhibitory activity than acarbose [34,35]. Two binding manners between enzymes and flavonoids have been detected: (i) flavonoids bind directly to amino acid residues in the active sites of enzymes and exclude the binding of substrate; (ii) flavonoids can interact with amino acid residues near the active site and close the channel to the active center [35].

### 3.3. DPPH Scavenging Effect of FFM

As a result of various endogenous systems and exposure to various physico-chemical conditions, or pathological states, our bodies produce reactive oxygen and reactive nitrogen species, known as radicals. For proper physiological function, a balance between antioxidants and free radicals is required. Oxidative stress occurs when free radicals overwhelm the body’s ability to regulate them. Free radicals alter lipids, proteins and DNA, resulting in a wide range of human diseases. Hence consumption of antioxidants’ rich diet can assist in coping with this oxidative stress [36,37]. Our results revealed that FFM extract displayed free radical scavenging character with IC_50_ values of 5.37 ± 0.41 µg/mL, being comparable with the standard antioxidant trolox (IC_50_ = 2.23 ± 1.23 µg/mL). Previous studies demonstrated that flavonoids and other polyphenols are able to scavenge the DPPH free radicals by donating their hydrogen [38]. This result is in accordance with that obtained in a study conducted by Sim et al., where he reported that kaempferol, luteolin, and myricetin displayed DPPH radical scavenging activities with IC_50_ values 12, 5, and 4 μM [39].

### 3.4. Anti-Lipase Activity

Obesity and overweight are chronic metabolic disorders caused by an imbalance of energy expenditure and intake. They are major risk factors for cancer as well as cardiovascular, metabolic and endocrine diseases [40]. The lipolytic pancreatic lipase enzyme is synthesized and secreted by the pancreas, plays a functional role in the digestion of fats and is responsible for the hydrolysis of 50–70% of total dietary lipids. Determination of the anti-lipase effects is one of the most widely investigated mechanisms for evaluating the efficiency of synthetic and natural products as anti-obesity medications [41]. The current study showed that the FFM has potent porcine pancreatic lipase inhibitory activity with an IC_50_ value of 39.81 ± 1.23 µg/mL, compared with that of orlistat (IC_50_ = 12.56 ± 0.35 µg/mL). Shimura et al. reported that kaempferol, luteolin and myricetin inhibited lipase activity with IC_50_ of 0.72, 2.03 and 2.54 µg/mL, respectively [42].

In fact, orlistat is a potent inhibitor of pancreatic lipase enzyme produced from *Streptomyces toxytricini* bacteria. This clinically approved medication is intended for the treatment of obesity and overweight by decreasing the digestion of lipids [43].

### 3.5. Antimicrobial Activity

The microdilution assay was utilized to assess the FFM’s antimicrobial activity. The results showed that FFM has potent antibacterial activity against the *K. pneumoniae* strain, with a MIC value of 0.135 ± 0.01 µg/mL, being comparable with the reference drug ciprofloxacin (MIC = 0.13 ± 0.02 µg/mL) and higher than ampicillin (MIC = 1 ± 0.02 µg/mL). Furthermore, FFM extract showed higher antibacterial activity against *S. aureus*, *E. coli* and MRSA with MIC values of 1.5 ± 0.07, 2.25 ± 0.07 and 25 ± 1.01µg/mL, respectively, than the reference drug ampicillin (MIC= 3.12 ± 0.02, 3.12 ± 0.23, and 60.5 ± 0.71 µg/mL, respectively). It exhibits antibacterial activity against *P. aeruginosa* with a MIC of 12.5 ± 0.98 µg, comparable to ampicillin. The FFM showed potent antifungal activity against *C. albicans*, compared with the commercial antifungal drug fluconazole with MIC values of 6.25 ± 0.48 and 1.56 ± 0.01 µg/mL, respectively. Finally, this plant fraction exhibited a weak anti-mold activity (*E. floccosum)* with a MIC value of 12.5 ± 0.88 µg /mL, compared with fluconazole, which was used as a positive control and has a MIC value of 0.78 µg /mL.

Obeidat demonstrated that *M. autumnalis* fruits’ ethanolic extract displayed potential antimicrobial activity against *E. coli*, *P. aeroginosa*, MRSA and *C. albicans* [44].

Extracts of numerous medicinal plants encompassing flavonoids have been previously stated to possess antimicrobial activity [45,46]. Flavonoids are well established as antimicrobial agents for a wide spectrum of microbial strains. However, with the rising incidence of microbial resistance to antibiotics, flavonoids have become fascinating targets due to their usefulness as powerful alternatives to antibiotics. The presence of hydroxyl groups at certain places in flavonoids structures, particularly on aromatic rings, promotes antibacterial action. Hydrophobic substituents such as alkyl chains, alkylamino chains, prenyl groups and oxygen or nitrogen-containing heterocyclic sections improved the antibacterial activity of all flavonoids [47]. Flavonoids have a variety of modes of action against microbial species, including porin inhibition on the cell membrane, adhesion and biofilm development, energy metabolism, cytoplasmic membrane function and nucleic acid synthesis. Furthermore, flavonoids can alter the permeability of microbial cell membranes and reduce pathogenicity [48].

To the best of the authors’ knowledge, this is the first study that has looked at the chemical content and biological activity of *M. autumnalis* fruits. More research is needed to establish these putative pharmacological characteristics of the FFM extract of *M. autumnalis* in order to assess its impact in animal models and build suitable pharmaceuticals or food supplement formulations from this fraction.

## 4. Materials and Methods

### 4.1. Chemicals, Reagents and Instruments

The rat L6 muscle cell line and stably expressing myc-tagged GLUT4 (L6-GLUT4myc cells) were obtained from (Kerafast, Boston, MA, USA). Fetal bovine serum, modified Eagle’s medium (*α*-MEM), standard culture medium and all of the other tissue culture reagents were purchased from (Biological Industries, Beit Haemek, Israel). Horseradish peroxidase (HRP)-conjugated goat anti-rabbit antibodies were obtained from (Promega, Madison, WI, USA). Polyclonal anti-myc (A-14), the 3-(4,5-dimethylthiazol-2-yl)-2,5-diphenyl tetrazolium bromide (MTT) reagent, methoxyamine hydrochloride, pyridine, *N*-methyl-*N*-(trimethylsilyl)-trifluoroacetamide (MSTFA) and other standard chemicals were purchased from (Sigma-Aldrich, St. Louis, MO, USA).

All of the solvents used were HPLC-grade and acquired from (Alfa Aesar, Heysham, Lancashire, England). Tris-HCl, silica gel 60 GF254 thin-layer chromatography, silica gel (100–200 and 200–300 mesh) for column chromatography, porcine pancreatic amylase and lipase enzymes were acquired from (Sigma-Aldrich, Burlington, MA, USA). Corn starch was purchased from (Alzahra Company, Nablus, Palestine). Ampicillin, 3,5-dinitrosalicylic acid, ciprofloxacin and orlistat were acquired from (Sigma-Aldrich, Anekal Taluk, Bangalore, India). Acarbose, *α*-glucosidase, phosphate buffer, Na_2_CO_3_, 2,2-diphenyl-1-picrylhydrazyl (DPPH) and trolox were purchased from Sigma-Aldrich (Taufkirchen, Germany). Finally, Mueller-Hinton broth, fluconazole, Sabouraud dextrose agar and dimethyl sulphoxide (DMSO) were acquired from Himedia (Mumbai, Maharashtra, India).

The ^1^H- and ^13^C-NMR spectra were obtained using a Bruker AV-400 instrument and DMSO-d_6_ as a solvent. Spectrophotometer (Jenway 7315, Staffordshire, England), microtiter plate reader (Anthos, USA), stir-mixer (Tuttnauer, Jerusalem, Palestine), weighing scale (Adam Equipment Inc., Oxford, CT, USA), multichannel micro-pipet (Eppendorf, Hamburg, Germany(, shaker incubator apparatus (Memmert incurboration, Büchenbach, Germany), sterile syringe filter hydrophobic (PTFE, Yangtze River Delta, China) and an incubator with CO_2_ (Tuttnauer, Jerusalem, Palestine) were used to complete the current study; the microplate reader was from Anthos, Biochrom (Cambridge, UK).

### 4.2. Plant Collection and Preparation

In May 2020, the golden mature fruits of *M. autumnalis* were harvested in the Nablus highlands of Palestine. The plant was validated in the Herbal Products Laboratory at An-Najah National University and deposited in the same laboratory under the voucher specimen code Pharm-PCT-1509. *M. autumnalis* fruits were cleaned with distilled water and dried in the shade at room temperature (25 ± 2 °C). After four weeks, the total dry fruits were finely crushed and kept in a tightly sealed glass jar for future use.

### 4.3. Extraction and Isolation of Compounds

*M. autumnalis* fruit powder (200 g) was macerated with methanol (0.5 L), and the suspension was agitated at 35 °C for 24 h. The resultant extract was filtered, and the residual solid was extracted again with 0.5 L of methanol under the same conditions. Methanol was removed under reduced pressure to offer dark green sticky oil. Water (50 mL) was added to the concentrated green extract before extracting it with pentane to remove lipids, waxes and nonpolar chemicals and ethyl acetate to extract the flavonoids and other polyphenols. The biological effects of the ethyl acetate fraction of the methanolic extract of *M. autumnalis* fruits were studied.

The ethyl acetate extract (3 g) was subjected to silica gel column chromatography utilizing gradient dichloromethane–EtOAc (10:0–1:1) to obtain four fractions. Fraction 2 (0.687 g) was subjected to silica gel column chromatography employing gradient elution (hexane: acetone 10:0–10:1) to give compound **1** (6 mg). Fraction 3 (0.58 g) was subjected to silica gel column chromatography eluted with hexane–acetone (95:5–9:10) to obtain two fractions. Fraction 3a was subjected repeatedly to silica gel column chromatography eluted with hexane–acetone (95:5) and was finally purified by a preparative thin-layer chromatography column eluted with CH_2_Cl_2_-EOAc (7:3) to give compound **2** (6 mg). Fraction 3b was subjected to silica gel column chromatography using gradient elution with hexane–acetone (99:1–90:10) followed by preparative TLC to give compound **4** (7.6 mg). Fraction 4 (0.72 g) was subjected to repeated silica gel column chromatography using gradient elution CH_2_Cl_2_-EtOAc (9:1–6:4) led to the isolation of compound **3** (4.3 mg) [23].

### 4.4. Biological Method

#### 4.4.1. Cell Growth and Treatment

Cells from the rat L6 muscle cell line, stably expressing myc-tagged GLUT4 (L6-GLUT4myc), obtained from Kerafast (Boston, MA, USA), were maintained under an atmosphere of 95% air and 5% CO_2_ in *α*-MEM supplemented with 10% fetal calf serum (FCS), 100 U/mL penicillin and 0.1 mg/mL streptomycin.

#### 4.4.2. Cytotoxicity

The cellular metabolic activity was assessed as an indication of cell viability using the MTT test methodology devised by [49]. This procedure relies on the reduction of the yellow-colored 3-(4,5-dimethylthiazol-2-yl)-2,5-diphenyl tetrazolium bromide (MTT) by cellular enzymes to the purple formazan. L6-GLUT4myc cells were seeded at a density of 2 × 10^4^ cells/well in 96-well plates with 100 µL of culture media in each well. After 24 h, FFM was administered at increasing doses (0–1 mg/mL) for 20 h. The cell media was then changed with 100 µL of new medium/well containing MTT (0.5 mg/mL) and cultured for 4 h in the cell’s incubator in the dark. The supernatant was withdrawn from each well, and 100 µL of isopropanol/HCl (1 mM HCl in 100% isopropanol) was added. Using a microtiter plate reader, the absorbance at 620 nm was measured. A microtiter plate reader was used to detect absorbance at 620 nm. To serve as controls, two wells on each plate were left unfilled. All of the experiments were conducted three times [50]. To illustrate the effect of the FFM on cell viability, the following formula was used:$$\mathrm{Percent}\;\mathrm{viability}=\left(\frac{A620\;\mathrm{nm}\;\mathrm{of}\;\mathrm{plant}\;\mathrm{extract}\;\mathrm{treated}\;\mathrm{sample}}{A620\;\mathrm{nm}\;\mathrm{of}\;\mathrm{nontreated}\;\mathrm{sample}}\right)\times 100$$

#### 4.4.3. Determination of Surface GLUT4myc

Surface myc-tagged GLUT-4 was assessed in intact, non-permeabilized cells using an anti-myc antibody followed by a secondary antibody coupled to horseradish peroxidase, as previously described [51]. Cells were seeded in 24-well plates for 24 h before the plant extracts were added for 20 h. The cells were then serum-starved for 3 h before being treated for 20 min with or without 100 nM insulin and rinsed twice with ice-cold PBS. The cells were fixed for 10 min in 3% paraformaldehyde, incubated for 10 min in 0.1 M glycine, blocked for 10 min in 3% (*v*/*v*) goat serum, then reacted with polyclonal anti-myc antibody (1:200) for 1 h at 4 °C before being washed 10 times in PBS. Following that, cells were incubated for 1 h at 4 °C with horseradish peroxidase-bound goat anti-rabbit secondary antibody (1:1000) before being washed 15 times with PBS. Cells were then treated with 1 mL of phenylenediamine dihydrochloride reagent and allowed to develop in the linear range for 20–30 min in the dark at room temperature. The reaction was quenched by the addition of 1 mL of 3N HCl to each well. Supernatants were collected, and absorbance at 492 nm was measured. All of the results were subtracted from the background absorbance obtained in the absence of anti-myc antibodies.

#### 4.4.4. α-Amylase Inhibitory Method

25 mg of FFM was dissolved in 10% DMSO (25 mL). The solution was diluted using 0.02 M sodium phosphate buffer (pH 6.9) to generate the following dilutions: 10, 50, 70, 100 and 50 μg/mL. In parallel, a working solution of the *α*-amylases enzyme (2 units/mL) was made by combining 12.5 mg of this enzyme powder with a few drops of 10% DMSO, and a buffer solution was added to achieve 100 mL. The starch solution was made by dissolving 1 g of corn starch in distilled water (100 mL). A combination of 200 μL of FFM stock solution, 200 μL of *α*-amylase stock solution and 200 μL of corn starch solution was incubated in a water bath at 30 °C for 15 min. In addition, 3,5-dinitrosalicylic acid was added to the mixture and cooked for 10 min in a water bath at 85–90 °C. After allowing the mixture to cool to room temperature, 5 mL of distilled water was added. The blank solution was created by substituting the plant portion with 200 μL of the buffer. Acarbose was employed as a positive control, and absorbance was measured at 540 nm using a Spectrophotometer (UV-Vis). The *α*-amylase inhibitory concentrations were estimated using the following equation:I (%)= [ABS_blank_ − ABS_test_]/[ABS_blank_]) × 100%
where I (%) is the α-amylase inhibitory percentage [52].

#### 4.4.5. α-Glucosidase Inhibitory Activity

The stock solution was prepared by dissolving a portion of the FFM (1 mg) in phosphate buffer (1 mL). The resulting solution was diluted with phosphate buffer to achieve different concentrations (100, 200, 300, 400 and 500 μg/mL). Twenty microliters of the stock solution and *α*-glucosidase solution (1 unit/mL) were then mixed with 50 μL of phosphate buffer and incubated in a water bath at 37 °C for 15 min. Thereafter, 20 μL of P-NPG solution was added to the mixture and incubated for 20 min at 37 °C before terminating the reaction with 50 μL of 0.1 M Na_2_CO_3_. The blank solution was prepared by replacing the FFM solution with phosphate buffer. Acarbose was employed as a positive control, and absorbance was measured at 405 nm using a UV-Vis spectrophotometer. The *α*-glucosidase inhibitory activity was calculated using the following formula:I (%) = [ABS_blank_ − ABS_test_]/[ABS_blank_]) × 100%
where I (%) is the percentage inhibition of *α*-glucosidase [53].

#### 4.4.6. DPPH Scavenging Activity

A 1 mg/mL working solution was made by dissolving 100 mg of FFM in 100 mL of methanol, which was then diluted with methanol to yield serial concentrations (2, 5, 10, 20, 50 and 100 μg/mL). Methanol (1 mL) was added to the FFM stock solution (1 mL), followed by DPPH solution (1 mL), and the combination was incubated for 30 min at room temperature under the light exclusion. The blank solution was prepared by replacing the plant fraction solution with methanol. Trolox was employed as a positive control, and absorbance was measured at 517 nm using a UV-Vis spectrophotometer and compared to the positive control. The DPPH scavenging activity was calculated using the following equation:I (%) = [ABS_blank_ − ABS_test_]/[ABS_blank_]) × 100%
where I (%) is the percentage of DPPH scavenging activity [54].

#### 4.4.7. Porcine Pancreatic Lipase Enzyme Inhibitory Assay

In a beaker, 100 mg of FFM was dissolved in 10% DMSO to achieve a series of concentrations (50, 100, 200, 300 and 400 µg/mL). Right before use, a lipase enzyme stock solution (1 mg/mL) was produced by dissolving lipase enzyme powder (25 mg) in 10% DMSO (25 mL). The p-nitrophenyl butyrate (PNPB) stock solution was prepared by dissolving PNPB (20.9 mg) in acetonitrile (2 mL). Then, 0.2 mL of each of the FFM serial dilutions was mixed with lipase enzyme stock solution (0.1 mL) and Tris-HCl solution was added to the mixture to bring the total volume to 1 mL. After 15 min of incubation at 37 °C in a water bath, a 100 μL of PNPB solution was added and the mixture was incubated for 30 min at 37°C. The blank solution was prepared by mixing 100 μL of the lipase enzyme solution (1 mg/mL) with a Tris-HCl solution (0.9 mL). Orlistat was used as a positive control. The absorbance was measured using a spectrophotometer (UV-Vis) at 405 nm. Lipase enzyme inhibitory percentage was calculated employing the next equation [41]:I (%) = [ABS_blank_ − ABS_test_]/[ABS_blank_]) × 100%.

#### 4.4.8. Antimicrobial Activity

The bacterial and fungal strains used in the current investigation were obtained from the American Type Culture Collection (ATCC) and clinical isolates. *Staphylococcus aureus* (ATCC 25923), *Enterococcus faecium* (ATCC 700221) and Methicillin-Resistant *Staphylococcus aureus* (clinical isolate) belong to Gram-positive strains. In addition to *Shigella sonnie* (ATCC 25931), *Pseudomonas aeruginosa* (ATCC 27853), *Klebsiella pneumoniae* (ATCC 13883) and *Escherichia coli* (ATCC 25922), which are classified as Gram-negative species.

The effect of the FFM against the fungal strains including *Epidermatophyton floccosum* (ATCC 10231) and *Candida albicans* (ATCC 90028) were also evaluated. The antimicrobial activity of the FFM against fungal and bacterial strains was evaluated using the microdilution technique. Mueller–Hinton broth (8.8 g) was autoclaved and kept at 4 °C after 400 mL of distilled water was added. The extract was first dissolved in sterile distilled water at a concentration of 1 mg/mL. In each well of the 96-well microdilution plate, 100 µL of the prepared broth was distributed; subsequently, serial dilution of the extract solution was performed obtaining concentrations ranging from 500 to 0.5 µg/mL. Well number 12 was left free from the plant material and was considered a control for microbial growth. A fresh bacterial colony was picked from an overnight agar culture and was prepared to match the turbidity of the 0.5 McFarland standard to provide a bacterial suspension of 1.5 × 10^8^ colony forming units (CFU)/mL. This suspension was diluted with broth by a ratio of 1:3 to reach 5 × 10^7^ CFU. Then, 1 µL of the bacterial suspension was added to all of the wells except well number 11, which was considered as a negative control for microbial growth. The plate was then incubated at 35 °C for 18 h.

The microdilution method for the yeast *C. albicans* was performed as described above, except that after matching the yeast suspension with the McFarland standard, which was diluted with NaCl (0.9%) by a ratio of 1:50, followed by 1:20, and 100 µL was placed in the wells. The plate was incubated for 48 h instead of 18 h.

The agar dilution method was carried out to investigate the antimold activity of FFM against *Epidermatophyton floccosum*. Sabouraud dextrose agar (1 mL) was placed in each test tube and kept at 40 °C in a water bath. Then, FFM (1 mL) was mixed with Sabouraud dextrose agar (1 mL) in the first tube and serial dilutions were performed in six tubes. Tube number 6 was kept free of any plant material and was considered as a negative control for microbial growth.

After the Sabouraud dextrose agar was solidified, the spores of the mold culture were dissolved in distilled water containing 0.05% Tween 80 and scratched from the plate for comparison with McFarland turbidity. Twenty microliters of the obtained suspension was pipetted into all of the tubes except tube number 5, which was considered as a negative control. The tubes were incubated at 25 °C for 14 days [55].

In the current antimicrobial experiments, ampicillin, ciprofloxacin and fluconazole, (powder purity ≥98%), which all obtained from Sigma-Aldrich (Germany), were utilized as positive controls and prepared via the above-mentioned method. Each experimental work was repeated in triplicates.

### 4.5. Statistical Characterizations

The conducted experiments were determined in triplicates. The results are presented as means ± standard deviation (SD). GLUT4-myc translocation results are presented as the mean ± SEM. The data were analyzed using *t*-test statistical calculations conducted with the SPSS version 21.0 software. When comparing samples, differences were considered statistically significant when *p* < 0.05.

## 5. Conclusions

The current investigation documented for the first time the chemical components of *M. autumnalis* ripe fruit flavonoid fraction and evaluated its antidiabetic, DPPH scavenging, antiobesity, cytotoxic and antimicrobial effects. Four flavonoids were isolated and characterized from the flavonoids fraction, namely kaempferol **1**, luteolin **2**, myricetin **3** and (+)-taxifolin **4**. The results indicate that FFM dramatically increased the amount of GLUT4 in the PM in both the presence and the absence of insulin. These findings suggest that FFM has anti-diabetic properties. Furthermore, FFM revealed *α*-amylase and *α*-glucosidase inhibitory characteristics and was shown to have strong DPPH scavenging and porcine pancreatic lipase inhibitory activities. In addition, FFM inhibited the growth of all of the tested bacterial and fungal strains and showed the greatest antibacterial activity against the *K. pneumoniae* strain. Moreover, the cytotoxicity screening showed that FFM is found to be safe up to 1 mg/mL. Future plans are required to confirm these very important outcomes by in vivo trials and to design suitable formulations to be used in the pharmaceutical and food supplement industries.

## Figures and Tables

**Figure 1 molecules-27-01046-f001:**
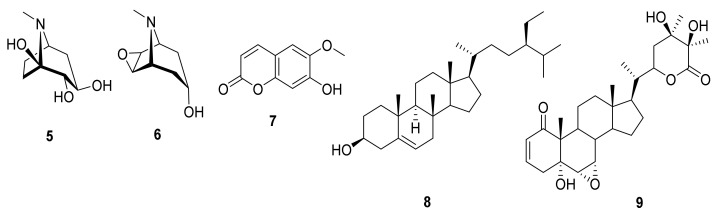
Phytochemicals isolated previously from *M. autumnalis*.

**Figure 2 molecules-27-01046-f002:**
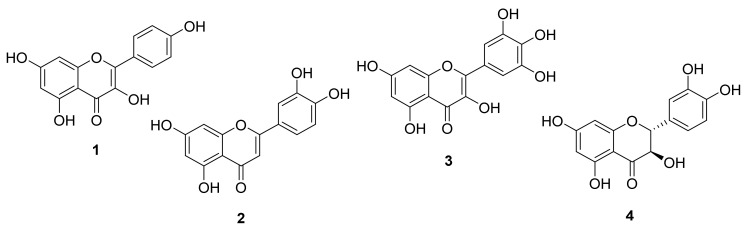
Chemical structures of the isolated compounds from FFM.

**Figure 3 molecules-27-01046-f003:**
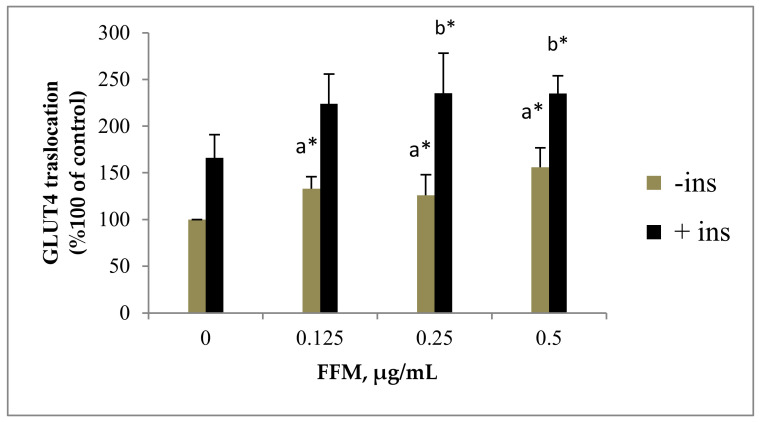
Effect of FFM on GLUT4 translocation. GLUT4 L6-GLUT4myc cells (100,000 cell/well) were exposed to FFM for 20 h. Serum-depleted cells were treated without (−) or with (+) 1 nM insulin for 20 min at 37 °C and surface myc-tagged GLUT4 density was quantified using the antibody-coupled colorimetric assay. Values given represent means ± SEM (% of untreated control cells) of three independent experiments carried out in triplicates. * *p* < 0.05; a: compared with − insulin control group, b: compared with + insulin control group.

**Figure 4 molecules-27-01046-f004:**
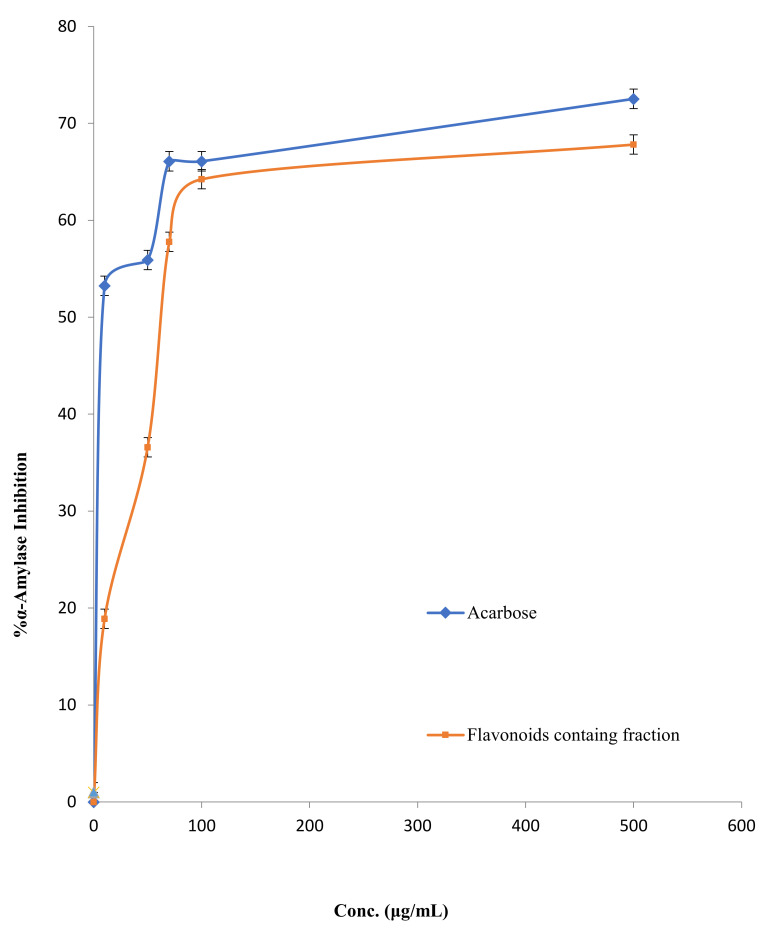
α-Amylase inhibitory activity of FFM and acarbose.

**Figure 5 molecules-27-01046-f005:**
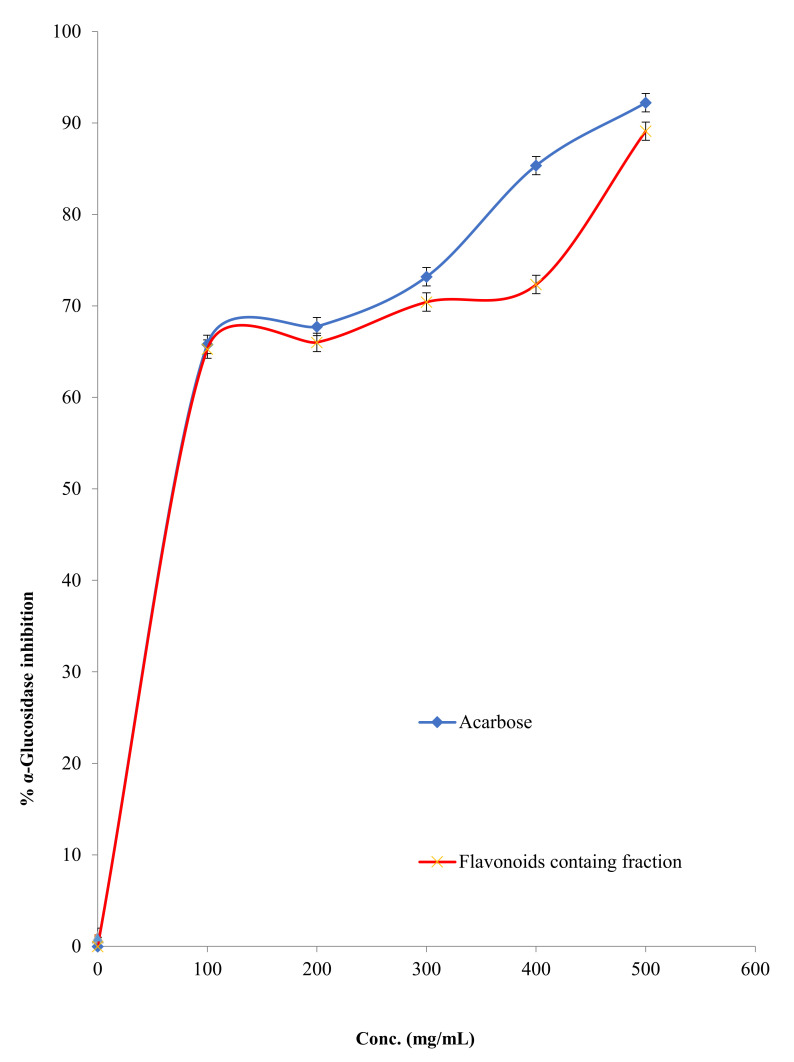
α-Glucosidase inhibition percentage of the FFM and acarbose.

**Figure 6 molecules-27-01046-f006:**
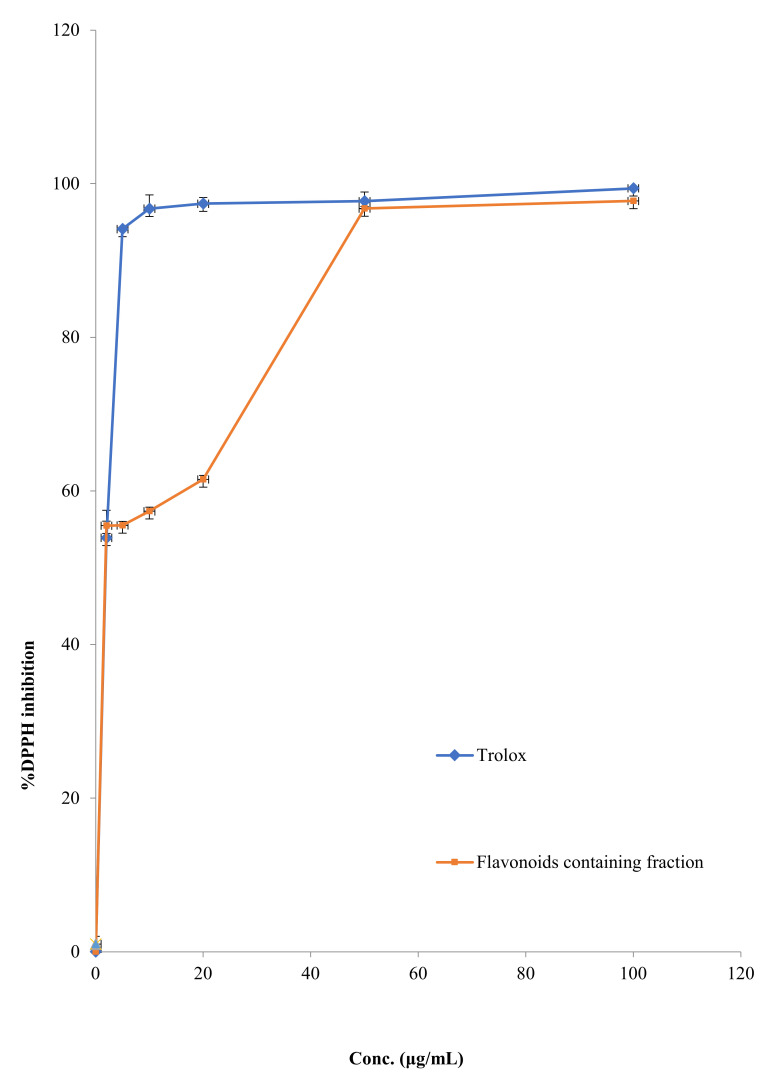
DPPH inhibitory potentials by FFM and trolox.

**Figure 7 molecules-27-01046-f007:**
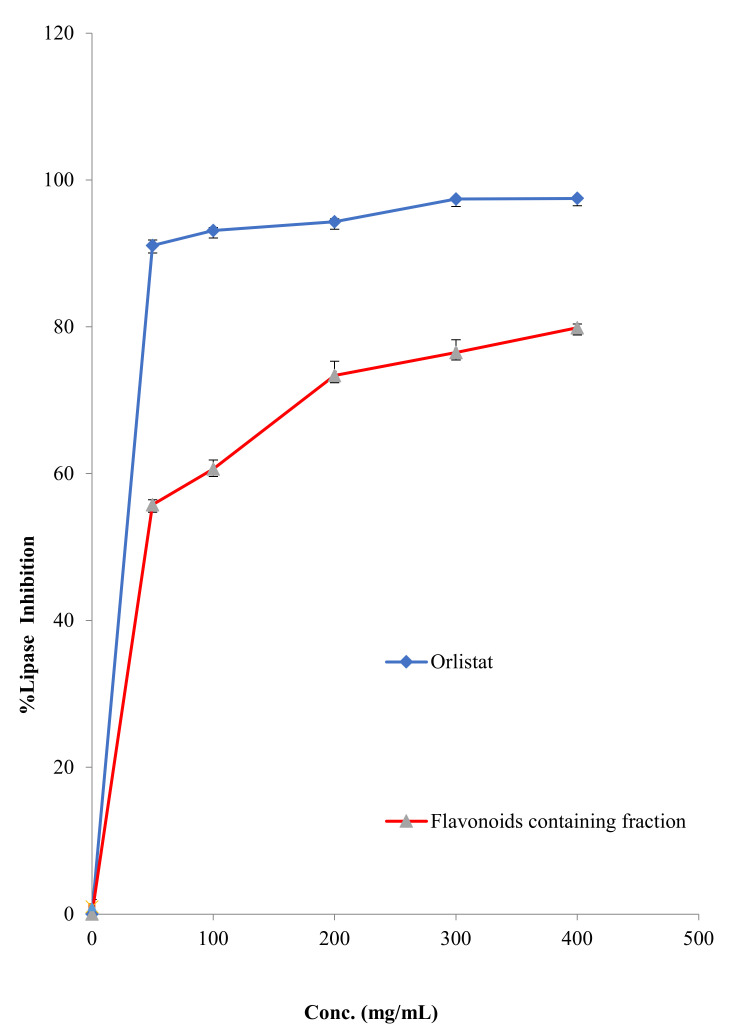
The lipase inhibition percentage of FFM and orlistat.

**Table 1 molecules-27-01046-t001:** ^1^H and ^13^C spectral data of **1**–**4** (DMSO-d_6_, 400 Hz, δ in ppm, J in Hz).

	Compound 1		Compound 2		Compound 3		Compound 4	
C No	*δ* _C_	*δ* _H_	*δ* _C_	*δ* _H_	*δ* _C_	*δ* _H_	*δ* _C_	*δ* _H_
1								
2	147.3		169.1		147.3		72.0	4.98 d, *J* = 11.2 Hz
3	136.1		108.1	6.68 s	136.3		83.5	4.50 d, *J* = 11.2, 6.2 Hz
4	176.4		186.9		176.2		197.3	
4a	103.5		108.9		103.4		101.0	
5	161.2		166.7		161.2		163.8	
6	98.7	6.19 d, *J* = 2.1 Hz	103.9	6.19 d, *J* = 2.1 Hz	98.6	6.18 d, *J* = 2.1 Hz	95.4	5.86 d, *J* = 2.1 Hz
7	164.4		169.3		164.3		167.3	
8	94.0	6.44 d, *J* = 2.1 Hz	99.4	6.45 d, *J* = 2.1 Hz	93.7	6.37 d, *J* = 2.1 Hz	96.4	5.91 d, *J* = 2.1 Hz
8a	159.67		162.2		156.5		163.0	
1′	122.1		126.6		121.2		128.5	
2′	130.0	8.05 d, *J* = 8.9 Hz	118.3	7.40 d, *J* = 2.2 Hz	107.6	7.24 s	115.8	6.88 d, *J* = 1.6 Hz
3′	115.9	6.93 d, *J* = 8.9 Hz	150.7		146.2		144.9	
4′	156.6		154.9		136.3		145.4	
5′	115.9	6.93 d, *J* = 8.9 Hz	120.9	6.89 d, *J* = 8.3 Hz	146.2		115.6	6.75 m, 2 H overlap with H-6′
6′	130.0	8.05 d, *J* = 8.9 Hz, 2 H	124.1	7.43 dd, *J* = 8.3, 2.2 Hz	107.6	7.24 s	119.9	6.75 m, overlap with H-5′
3-OH		10.80				10.79		5.77 d, *J* = 6.2 Hz
5-OH		12.49		12.99		12.51		11.92
7-OH		10.12		10.85		9.36		10.85
4′-OH		9.42		9.95		8.82		9.05
3′-OH				9.43		9.23		9.00
5′-OH						9.23		

(+)-taxifolin (**4**): Pale yellow solid, [α]^21^ + 52.3 acetone (c = 0.25, acetone).

**Table 2 molecules-27-01046-t002:** MIC values (µg/mL) of FFM and positive controls.

Microbial Strains	Fluconazole	Ampicillin	Ciprofloxacin	FFM
*S. aureus*	0	3.12 ± 0.02	0.78 ± 0.01	1.5 ± 0.07
*E. faecium*	0	1.56 ± 0.01	0.78 ± 0.02	3.125 ± 0.03
*E. coli*	0	3.12 ± 0.23	1.56 ± 0.08	2.25 ± 0.07
*P. aeruginosa*	0	12.5 ± 0.13	3.12 ± 0.11	12.5 ± 0.98
*K. pneumoniae*	0	1 ± 0.02	0.13 ± 0.02	0.135 ± 0.01
*P. vulgaris*	0	18 ± 1.05	15 ± 0.35	22 ± 0.97
MRSA	0	60.5 ± 0.71	12.5 ± 0.54	25 ± 1.01
*E. floccosum*	0.78 ± 0.01	0	0	12.5 ± 0.88
*C. albicans*	1.56 ± 0.01	0	0	6.25 ± 0.48

## Data Availability

All data is contained within the article.

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
