# Peer review of "Isolation, Identification and Pharmacological Effects of Mandragora autumnalis Fruit Flavonoids Fraction"

_molecules, 2022, doi:10.3390/molecules27031046_

Round 1
Reviewer 1 Report
The manuscript is dealing with the extraction and characterization of the medicinal plant Mandragora officinalis Mill.
The authors used a variety of analytical methods to prove their findings.
The antimicrobial activity was tested on different strains successfully. The plant, with its major constituents, the flavonoids, seems to be a successful candidate for further use in pharmaceutical products.
Few remarks:
1-The introduction should include few sentences about the plant bio-components, their potential uses and the antibiotic resistance problem. The latter is mentioned in one sentence without reference (316-317). This can be added to the introduction to improve the manuscript further.
2-The strains could have been discussed in more systematic way. MRSA is mentioned and portrayed as ATCC25923 (497-500). The sentences used are too long, confusing and need to be corrected.
3-The discussion of the antimicrobial activities needs further work. This will allow to clearly state the potential use of Mandragora officinalis Mill. in the conclusions.
4-The reference section needs a few corrections related to the volume to make it MDPI format
Good luck
Author Response
Reviewer 1
The manuscript is dealing with the extraction and characterization of the medicinal plant Mandragora officinalis Mill.
The authors used a variety of analytical methods to prove their findings.
The antimicrobial activity was tested on different strains successfully. The plant, with its major constituents, the flavonoids, seems to be a successful candidate for further use in pharmaceutical products.
Few remarks:
1-The introduction should include few sentences about the plant bio-components, their potential uses and the antibiotic resistance problem.
Thank you for these important suggestions and now we added plant bio-components, the evidence based uses of the plant and a paragraph about antibiotic resistance problem.
The latter is mentioned in one sentence without reference (316-317). This can be added to the introduction to improve the manuscript further.
We moved this sentence to the introduction part as you suggested and we added suitable citation.
2-The strains could have been discussed in more systematic way. MRSA is mentioned and portrayed as ATCC25923 (497-500). The sentences used are too long, confusing and need to be corrected.
Thank you for this advice and now we corrected this whole paragraph.
3-The discussion of the antimicrobial activities needs further work. This will allow to clearly state the potential use of Mandragora officinalis Mill. in the conclusions.
We improved the antimicrobial part discussion and conclusion according to your advice and compared our antimicrobial results with previously conducted studies. Thank you
4-The reference section needs a few corrections related to the volume to make it MDPI format
We corrected and revised the whole references part.
We would like to thank the respected reviewer for the valuable time spent reviewing our manuscript and the important comments he has made.
Reviewer 2 Report
The paper describes isolation of four known flavonoids, two of them of common occurrence, from the ethyl acetate fraction of methanolic extract from ripe, dried fruits of mandrake. Cytotoxic, antiradical (DPPH) and antimicrobial activities of the ethyl acetate fraction as well as its inhibitory effect on α-amylase, α-glucosidase, porcine pancreatic lipase and effect on hexose transporter GLUT4 translocation to the plasma membrane were assessed.
The only aspect of novelty in the study is the source of the extract – mandrake fruits.
The assessed activities were repeatedly assayed for the flavonoids of different structures.
Although the activity profile of the studied fraction of the extract is interesting, in fact we do not know much about its chemical composition. The Authors obtained 3 g of the ethyl acetate extract and after the chromatographic separation managed to isolate: 6 mg of kaempferol, 6 mg of luteolin, 7.6 mg of (+)-taxifolin, and 4.3 mg of myricetin. The isolation yield was 0.8%. In this context, naming the ethyl acetate fraction of the extract “flavonoid fraction” seems to be a bit exaggerated. What about the remaining part of the fraction? Are the Authors sure that the fraction does not contain more active compounds?
In my opinion, the manuscript may be reconsidered for publication in “Molecules” after in-depth improvement.
Detailed comments:
Abstract, line 30 – “…was significantly observed…”. The effect was significant not the observation.
Abstract, line 31 – the enzyme inhibitory activities were not “potential”, they were real. Did the Authors mean “potent”?
Introduction, line 51 – “Mandragora officinalis Mill.” was the plant name present in “The Plant List” which is no longer valid. “The World Flora Online” (the current working list of plant species, successor of “The Plant List”) does not mention this name. Mandragora officinarum L. and Mandragora autumnalis Bertol. are the proper names.
Introduction, line 56 – “…to treat lung cancer and diabetes [4,5]”. None of the references contain information about the use of the plant in the diabetes treatment.
Introduction, line 64 – atropine is not the best example. Atropine is formed by racemization during the drying of the plant material and extraction procedure.
Line 65 – “…in various parts…” -> from various parts. Is sitosterol the most important steroid constituent of mandrake? What about withanolides?
Line 71 -.”…communicable diseases such as microbial resistance…”. This phrase needs rethinking and change.
Lines 80-81 – “…translocatase…” -> translocase
Results, line 114 – “…significantly observed…”. Please correct.
Results, lines 143-150 – instead of “antioxidant activity” DPPH scavenging activity should be used. More than one assay (assays based on different mechanisms) is needed to assess antioxidant potential of the extract.
Discussion, line 189 – “flavonoids extract” -> methanolic extract
Line 196 – [21], line 226 – [22], line 233 –[17], line 241 – [18], and so one. Erroneous numeration of references
Lines 242-244 – “Insulin-dependent glucose metabolism principally occurs in the hippocampus, a process that is mediated by GLUT4”. The Authors need to rethink and correct the sentence.
Lines 255-256 – “Kaempferol and Myricetin” -> kaempferol and myricetin.
Line 257 – [27,28] – erroneous numeration.
Lines 257 and 260 – “Kaempferol” -> kaempferol.
Line 263 – “potential”? activity.
Line 286 – “…and hence avoiding diseases.” The whole sentence needs correction.
Lines 296-297 – The sentence should be changed.
Line 309 – not nanoM but microM (erroneous units). Please recalculate units to make the comparison easy (microg/mL or microM).
Lines 310-311 – Are you sure that your fraction is more active than pure flavonoids?
Line 319 – “potential”? antibacterial activity.
Lines 338-339 – “Flavonoids have a variety of modes of action against antimicrobial agents,“. This would be unwanted activity. Please correct.
Materials and Methods, line 381 – “25±2” -> 25 ± 2.
Line 390 – “flavonoid fraction” -> ethyl acetate fraction of the methanolic extract.
Lines 392-402 – Please correct numeration of fractions. The reference 38 is redundant.
Line 413 – The reference 40 is erroneously placed.
Lines 420-424 – Absorbance was measured at 570 nm. Why the calculations were done for the absorbance measured at 620 nm?
Line 466 – Acarbose was used as a negative control?
Line 471 – “Antioxidant activity” -> DPPH scavenging activity.
Lines 497-500 – The bacterial and fungal strains were NOT obtained from selected species of methicillin-resistant Staphylococcus aureus (MRSA). The whole sentence needs correction.
Lines 506-513 – Repetition. Something went wrong with “copy and paste”.
Lines 508 and 509 – 100 L?
Line 512 – 100 microL of the FFM (the fraction was dissolved in buffer? Water? Alcohol? Please specify. This is essential for the interpretation of the results). An information concerning solubilization of the standard antibiotics is also necessary (lines 536-538).
Line 538 – The reference 45 is erroneously placed.
References, line 659 – year of publication is missing.
Figures 5-8 – error bars are missing
Table 1 – What about statistics?
Author Response
The paper describes isolation of four known flavonoids, two of them of common occurrence, from the ethyl acetate fraction of methanolic extract from ripe, dried fruits of mandrake. Cytotoxic, antiradical (DPPH) and antimicrobial activities of the ethyl acetate fraction as well as its inhibitory effect on α-amylase, α-glucosidase, porcine pancreatic lipase and effect on hexose transporter GLUT4 translocation to the plasma membrane were assessed.
The only aspect of novelty in the study is the source of the extract – mandrake fruits.
The assessed activities were repeatedly assayed for the flavonoids of different structures. Although the activity profile of the studied fraction of the extract is interesting, in fact we do not know much about its chemical composition.
We would like to thank the respected reviewer for appreciating the value of our work.
The Authors obtained 3 g of the ethyl acetate extract and after the chromatographic separation managed to isolate: 6 mg of kaempferol, 6 mg of luteolin, 7.6 mg of (+)-taxifolin, and 4.3 mg of myricetin. The isolation yield was 0.8%. In this context, naming the ethyl acetate fraction of the extract “flavonoid fraction” seems to be a bit exaggerated. What about the remaining part of the fraction? Are the Authors sure that the fraction does not contain more active compounds?
I completely agree with you. Because this fraction contains a lot of compounds as seen in the TLC, it must be called Flavonoids containing fraction rather than Flavoinds fraction. Some of the spots visualized by UV light, while others did not (staining with potassium permanganate and Ferric chloride stains). Furthermore, a few yellow and green spots were observed. The fraction contains various polyphenols and other compounds. The purified amounts do not imply that the isolated compounds are only present in these amounts in the ethyl acetate fraction. Plenty of material, especially polar compounds were lost during the repeated silica gel chromatography. We were unable to investigate the flavonoids glycosides due to a lack of preparative HPLC and reversed phase silica.
In my opinion, the manuscript may be reconsidered for publication in “Molecules” after in-depth improvement.
Detailed comments:
Abstract, line 30 – “…was significantly observed…”. The effect was significant not the observation.
Thank you for your efforts in improving our manuscript and now we corrected this sentence
Abstract, line 31 – the enzyme inhibitory activities were not “potential”, they were real. Did the Authors mean “potent”?
Corrected
Introduction, line 51 – “Mandragora officinalis Mill.” was the plant name present in “The Plant List” which is no longer valid. “The World Flora Online” (the current working list of plant species, successor of “The Plant List”) does not mention this name. Mandragora officinarum L. and Mandragora autumnalis Bertol. are the proper names.
Done and changed to be Mandragora autumnalis Bertol. Thank you for this advice and for the excellent website
Introduction, line 56 – “…to treat lung cancer and diabetes [4,5]”. None of the references contain information about the use of the plant in the diabetes treatment.
We corrected this sentence and refernces.
Introduction, line 64 – atropine is not the best example. Atropine is formed by racemization during the drying of the plant material and extraction procedure.
We deleted atropine
Line 65 – “…in various parts…” -> from various parts. Is sitosterol the most important steroid constituent of mandrake? What about withanolides?
Sitosterol is just one example of a steroid, and it is not the most important one. We added plant-isolated withanolides with adequate references
Line 71 -.”…communicable diseases such as microbial resistance…”. This phrase needs rethinking and change.
We deleted this sentence and we added new paragraph about AMR
Lines 80-81 – “…translocatase…” -> translocase
Corrected thank you
Results, line 114 – “…significantly observed…”. Please correct.
Done as requested
Results, lines 143-150 – instead of “antioxidant activity” DPPH scavenging activity should be used. More than one assay (assays based on different mechanisms) is needed to assess antioxidant potential of the extract.
Corrected throughout the whole manuscript.
Discussion, line 189 – “flavonoids extract” -> methanolic extract
Corrected
Line 196 – [21], line 226 – [22], line 233 –[17], line 241 – [18], and so one. Erroneous numeration of references
All these citations are now corrected, thank you
Lines 242-244 – “Insulin-dependent glucose metabolism principally occurs in the hippocampus, a process that is mediated by GLUT4”. The Authors need to rethink and correct the sentence.
This sentence now rephrased and corrected
Lines 255-256 – “Kaempferol and Myricetin” -> kaempferol and myricetin.
Corrected as requested
Line 257 – [27,28] – erroneous numeration.
Corrected as requested
Lines 257 and 260 – “Kaempferol” -> kaempferol.
Corrected as requested
Line 263 – “potential”? activity.
Corrected as requested
Line 286 – “…and hence avoiding diseases.” The whole sentence needs correction.
Corrected as requested
Lines 296-297 – The sentence should be changed.
We deleted this sentence as we added this recommendation in the conclusion section.
Line 309 – not nanoM but microM (erroneous units). Please recalculate units to make the comparison easy (microg/mL or microM).
Done
Lines 310-311 – Are you sure that your fraction is more active than pure flavonoids?
Corrected as requested
Line 319 – “potential”? antibacterial activity.
Corrected as requested
Lines 338-339 – “Flavonoids have a variety of modes of action against antimicrobial agents,“. This would be unwanted activity. Please correct.
Corrected
Materials and Methods, line 381 – “25±2” -> 25 ± 2.
Corrected
Line 390 – “flavonoid fraction” -> ethyl acetate fraction of the methanolic extract.
Corrected
Lines 392-402 – Please correct numeration of fractions. The reference 38 is redundant.
Done. Reference 38 has been relocated
Line 413 – The reference 40 is erroneously placed.
We deleted it as the methods cited in the beginning of 4.4.2. Cytotoxicity section
Lines 420-424 – Absorbance was measured at 570 nm. Why the calculations were done for the absorbance measured at 620 nm?
It was 620, we corrected it
Line 466 – Acarbose was used as a negative control?
Corrected
Line 471 – “Antioxidant activity” -> DPPH scavenging activity.
Done
Lines 497-500 – The bacterial and fungal strains were NOT obtained from selected species of methicillin-resistant Staphylococcus aureus (MRSA). The whole sentence needs correction.
Done
Lines 506-513 – Repetition. Something went wrong with “copy and paste”.
Corrected
Lines 508 and 509 – 100 L?
Corrected
Line 512 – 100 microL of the FFM (the fraction was dissolved in buffer? Water? Alcohol? Please specify. This is essential for the interpretation of the results). An information concerning solubilization of the standard antibiotics is also necessary (lines 536-538).
Thank you again and now we explain this point.
Line 538 – The reference 45 is erroneously placed.
Corrected
References, line 659 – year of publication is missing.
Corrected
Figures 5-8 – error bars are missing
They present in the figures but the SD values are very small
Table 1 – What about statistics?
Corrected
We would like to thank the respected reviewer for the valuable time spent reviewing our manuscript and the important comments he has made.
Reviewer 3 Report
This manuscript treats effects of a fraction named FFM obtained from Mandragora officinalis fruits containing several flavonoid aglycones. Although the findings shown seem to be valuable, the following points should be considered by the authors.
- Participation of the isolated flavonoids in the pharmacological effects of FFM should be estimated. Quantitative data for respective flavonoids in FFM are required for the explanation of the roles.
- Isolation yields seem to be too small for the explanation of the pharmacological effects of FFM with the flavonoid aglycones. Since many flavonoids have been found in plant fractions as glycosides, the authors should mention the relations of the flavonoid aglycones and glycosides in the FFM fraction.
- The authors should use bar graphs instead of the line graphs in Figures 5 – 8. Figure 3 is not required, and it can be replaced with sentences.
Author Response
This manuscript treats effects of a fraction named FFM obtained from Mandragora officinalis fruits containing several flavonoid aglycones. Although the findings shown seem to be valuable, the following points should be considered by the authors.
- Participation of the isolated flavonoids in the pharmacological effects of FFM should be estimated. Quantitative data for respective flavonoids in FFM are required for the explanation of the roles.
We were referring to the Flavonoids containing fraction (FFM). The biological activity of the ethyl extract is attributed to flavonoids and other biologically active extract compounds. Unfortunately, we did not perform the quantification, but we were tracking our target compounds (flavonoids) by staining all fractions with Ammonium phosphomolybdate and FeCl3 stains. Our goal was to isolate flavonoids from plants.
Isolation yields seem to be too small for the explanation of the pharmacological effects of FFM with the flavonoid aglycones. Since many flavonoids have been found in plant fractions as glycosides, the authors should mention the relations of the flavonoid aglycones and glycosides in the FFM fraction.
We know that most flavonoids in plants exist as flavonoid glycosides to increase their water solubility. We were unable to investigate the flavonoids glycosides due to a lack of preparative HPLC and reversed phase silica. We concentrated on the ethyl acetate extract because it contains some of the flavonoid aglycons that can be purified using silica gel chromatography. Flavonoids-containing fractions are referred to as FFM fractions. Our long-term goal is to investigate methanol extracts of plants containing flavonoids.
- The authors should use bar graphs instead of the line graphs in Figures 5 – 8.
We construct line graphs to obtain equations from which calculated IC50 values
- Figure 3 is not required, and it can be replaced with sentences.
Figure 3 deleted now and thank you for all these suggestions and advices
We would like to thank the respected reviewer for the valuable time spent reviewing our manuscript and the important comments he has made.
Once again, we would like to thank reviewers for the time and expertise in providing feedback. We look forward to hearing from you soon.
Kind regards,
Round 2
Reviewer 1 Report
Dear authors.
all is well except few flaws in the reference section.
I highlighted the recurring mistakes few times only. Please do the rest.
Wish you the best in further publications.
Author Response
Dear Reviewer
We would like to thank you for your constructive comments, which helped to improve our manuscript significantly. W
Please find our response to the reviewers' comments below. We carefully considered all of the comments and made the necessary changes to the manuscript. Throughout the manuscript, the revised text is highlighted in red font.
Dear authors.
Reviewer 1
all is well except few flaws in the reference section.
I highlighted the recurring mistakes few times only. Please do the rest.
Done
Wish you the best in further publications.
Thanks for help
Reviewer 2 Report
I would rather prefer to wait longer and get carefully revised manuscript. The present version (second one) definitely needs further improvement.
General comments:
The active fraction is of undefinite composition. Usually, at least chromatogram with description of the main signal(s) is required.
Taking into consideration metodology of MIC estimation, the values of standard deviation should be 0.0 or much higher than those given by the Authors. Unless the Authors used unconventional metod to calculate MIC. The method should be described in detail.
Antibiotics used as standards in MIC estimation should be described in Materials and Methods section in detail (manufacturer, purity, form).
The Authors claim that the values of standard deviation (or standard errors) were too small to be clearly visible in Figures 5-8. That raises the question: Are the results calculated for the three independent samples (three extracts prepared from the separated batches of the plant material) or they are simply three technical replications obtained using the same sample? This question should be clearly stated.
Detailed comments:
Figure legends – „Flavonoid fraction”. We agreed that the tested fraction of extract is not a flavonoid fraction. Please correct.
Page 2, lines 72-74 should be moved up the text (line 56) and enumeration of references should be changed accordingly.
Page 4, line 131 – Erroneous Figures numbering (two figures No 2).
Page 6, line 158 – Neither DPPH nor trolox is an enzyme.
Page 8, line 208 – Kaempferol -> kaempferol.
Page 8, line 219 – Reference No 22 does not contain NMR data and can not be used to support identification of the compound.
Page 9, line 254 – Erroneous numbering of references.
Page 11, lines 241-244 – Obeidat gives also MIC values (more useful to compare the results than the inhibition zone diameters).
Page 11, lines 344-347 – Information concerning silver particles is irrelevant to the present study.
Page 13, lines 438-442 – Again: 620 or 570 nm?
Page 13, line 440 – Erroneously placed reference.
Page 14, line 523 – The microbial activity...
Page 14, lines 527-529 need correction. Please specify the concentrations tested.
Author Response
We would like to thank you for your constructive comments, which helped to improve our manuscript significantly.
Please find our response to the reviewers' comments below. We carefully considered all of the comments and made the necessary changes to the manuscript. Throughout the manuscript, the revised text is highlighted in red font.
Reviewer 2
I would rather prefer to wait longer and get carefully revised manuscript. The present version (second one) definitely needs further improvement.
General comments:
The active fraction is of undefinite composition. Usually, at least chromatogram with description of the main signal(s) is required.
Yes, as indicated by tlc, the fraction contains a large number of compounds, including flavonoids (by staining with FeCl3). The majority of the nonpolar compounds were eliminated. We are not claiming that the ethyl acetate fraction of the metabolic extract contains only flavonoids. We referred to it as the flavonoids-containing fraction rather than the flavonoids-containing fraction.
Because the analytical HPLC is not attached to MS, the HPLC chromatogram is completely useless. We didn't have any standards because we didn't have a budget for the project. W wish we could use LC-MS, but unfortunately we don’t have this facility in Palestine. I hope you will contact me via email after we have completed our work so that we can collaborate in the future.
Taking into consideration metodology of MIC estimation, the values of standard deviation should be 0.0 or much higher than those given by the Authors. Unless the Authors used unconventional metod to calculate MIC. The method should be described in detail.
The microdillution experimental antimicrobial assay was conducted in triplicates. Low standard deviation means data are clustered around the mean, and high standard deviation indicates data are more spread out. The obtained data had a low standard deviation simply means that the data did not have a lot of “spread”. The MIC values in the sample didn't vary much, and are very tightly clustered together. You know sample means is got by averaging all sample values. So that standard variation also means the difference between sample values. Then low means this difference is low. Microdilution assay is a standardised method described by the Clinical and Laboratory Standards Institute (CLSI) which provide accurate detection so the obtained values from the triplicate experiments were relatively close.
Antibiotics used as standards in MIC estimation should be described in Materials and Methods section in detail (manufacturer, purity, form).
Done
The Authors claim that the values of standard deviation (or standard errors) were too small to be clearly visible in Figures 5-8. That raises the question: Are the results calculated for the three independent samples (three extracts prepared from the separated batches of the plant material) or they are simply three technical replications obtained using the same sample? This question should be clearly stated.
The results calculated for the three independent samples (three extracts prepared from the separated batches of the plant material
Detailed comments:
Figure legends – „Flavonoid fraction”. We agreed that the tested fraction of extract is not a flavonoid fraction. Please correct.
DONE
Page 2, lines 72-74 should be moved up the text (line 56) and enumeration of references should be changed accordingly.
Done
Page 4, line 131 – Erroneous Figures numbering (two figures No 2). Done
Page 6, line 158 – Neither DPPH nor trolox is an enzyme. Sorry for that. Done
Page 8, line 208 – Kaempferol -> kaempferol. Done
Page 8, line 219 – Reference No 22 does not contain NMR data and can not be used to support identification of the compound.
Done. I am sorry, it seems that I have to check all the references after the use of endnote
Page 9, line 254 – Erroneous numbering of references.
Done.
Page 11, lines 241-244 – Obeidat gives also MIC values (more useful to compare the results than the inhibition zone diameters).
We also removed the IZD because we couldn't compare the results to our previous research because our methods were different.
Page 11, lines 344-347 – Information concerning silver particles is irrelevant to the present study. Deleted
Page 13, lines 438-442 – Again: 620 or 570 nm? done
Page 13, line 440 – Erroneously placed reference. done
Page 14, line 523 – The microbial activity... Done
Page 14, lines 527-529 need correction. Please specify the concentrations tested. Done
Reviewer 3 Report
The manuscript was improved after considering the comments by this reviewer, and this reviewer think it is now acceptable for publication.
Author Response
Dear Reviewer,
Thank you for help
Best wishes
Nawaf